# Nebulized Colistin in Ventilator-Associated Pneumonia and Tracheobronchitis: Historical Background, Pharmacokinetics and Perspectives

**DOI:** 10.3390/microorganisms9061154

**Published:** 2021-05-27

**Authors:** Yinggang Zhu, Antoine Monsel, Jason A. Roberts, Konstantinos Pontikis, Olivier Mimoz, Jordi Rello, Jieming Qu, Jean-Jacques Rouby

**Affiliations:** 1Department of Pulmonary and Critical Care Medicine, Hua-Dong Hospital, Fudan University, Shanghai 200433, China; robinzyg@gmail.com; 2Multidisciplinary Intensive Care Unit, Department of Anaesthesiology and Critical Care, La Pitié-Salpêtrière Hospital, Assistance Publique Hôpitaux de Paris, Medicine Sorbonne University, 75012 Paris, France; antoine.monsel@aphp.fr; 3Unité Mixte de Recherche (UMR)-S 959, Immunology-Immunopathology-Immunotherapy (I3), Institut National de la Santé et de la Recherche Médicale (INSERM), 75012 Paris, France; 4Biotherapy (CIC-BTi) and Inflammation-Immunopathology-Biotherapy Department (DHU i2B), Hôpital Pitié-Salpêtrière, Assistance Publique-Hôpitaux de Paris, 75012 Paris, France; j.roberts2@uq.edu.au; 5University of Queensland Centre for Clinical Research, Faculty of Medicine The University of Queensland, 4006 Brisbane, Australia; 6Departments of Pharmacy and Intensive Care Medicine, Royal Brisbane and Women’s Hospital, 4006 Brisbane, Australia; 7Division of Anaesthesiology Critical Care Emergency and Pain Medicine, Nîmes University Hospital, University of Montpellier, 30029 Nîmes, France; 8Intensive Care Unit, First Department of Respiratory Medicine, School of Medicine, Sotiria General Hospital, National and Kapodistrian University of Athens, 15772 Athens, Greece; kostis_pontikis@yahoo.gr; 9Anaesthesiology and Intensive Care Department, University Hospital of Poitiers, University of Poitiers, 86000 Poitiers, France; o.mimoz@chu-poitiers.fr; 10Centro de Investigación Biomédica en Red de Enfermedades Respiratorias (CIBERES), Instituto de Salud Carlos III, 28029 Madrid, Spain; jrello@crips.es; 11Research & Innovation in Pneumonia & Sepsis, Vall d’Hebron Institute of Research (VHIR), 08035 Barcelona, Spain; 12Clinical Research, CHU Nîmes, Université Montpellier-Nîmes, 30029 Nîmes, France; 13Department of Pulmonary and Critical Care Medicine, Rui-jin Hospital, Shanghai Jiao-tong University School of Medicine, Shanghai 200127, China; jmqu0906@163.com; 14Institute of Respiratory Disease, Shanghai Jiao-tong University School of Medicine, Shanghai 200127, China

**Keywords:** nebulized polymyxin, nebulized colistimethate sodium, colistin, multidrug resistant gram-negative bacteria, ventilator-associated pneumonia, ventilator-associated tracheobronchitis, polylyxin resistance, phramacokinetic, pharmacodynamics, technique of nebulization

## Abstract

Clinical evidence suggests that nebulized colistimethate sodium (CMS) has benefits for treating lower respiratory tract infections caused by multidrug-resistant Gram-negative bacteria (GNB). Colistin is positively charged, while CMS is negatively charged, and both have a high molecular mass and are hydrophilic. These physico-chemical characteristics impair crossing of the alveolo-capillary membrane but enable the disruption of the bacterial wall of GNB and the aggregation of the circulating lipopolysaccharide. Intravenous CMS is rapidly cleared by glomerular filtration and tubular excretion, and 20–25% is spontaneously hydrolyzed to colistin. Urine colistin is substantially reabsorbed by tubular cells and eliminated by biliary excretion. Colistin is a concentration-dependent antibiotic with post-antibiotic and inoculum effects. As CMS conversion to colistin is slower than its renal clearance, intravenous administration can lead to low plasma and lung colistin concentrations that risk treatment failure. Following nebulization of high doses, colistin (200,000 international units/24h) lung tissue concentrations are > five times minimum inhibitory concentration (MIC) of GNB in regions with multiple foci of bronchopneumonia and in the range of MIC breakpoints in regions with confluent pneumonia. Future research should include: (1) experimental studies using lung microdialysis to assess the PK/PD in the interstitial fluid of the lung following nebulization of high doses of colistin; (2) superiority multicenter randomized controlled trials comparing nebulized and intravenous CMS in patients with pandrug-resistant GNB ventilator-associated pneumonia and ventilator-associated tracheobronchitis; (3) non-inferiority multicenter randomized controlled trials comparing nebulized CMS to intravenous new cephalosporines/ß-lactamase inhibitors in patients with extensive drug-resistant GNB ventilator-associated pneumonia and ventilator-associated tracheobronchitis.

## 1. Introduction

Polymyxins are non-ribosomal, cyclic oligopeptide antimicrobials, produced by the Gram-positive, spore-forming rod *Bacillus aerosporus* that were identified in 1946 from the soil of market gardens in England [1]. Among the five antibiotics that belong to the polymyxin group, only two can be used in human and veterinary medicine: polymyxin B (PMB) and polymyxin E, also called colistin. Colistin sulfate is available for oral and topical use. PMB sulfate and colistin methanesulfonate sodium (CMS) are available for aerosol use and intravenous administration. PMB sulfate—available in North and South America, Southeast Asia and Japan—combines two components with direct antibacterial activity: PMB1 and PMB2. CMS is an essentially inactive prodrug that is hydrolyzed to multiple components with direct bactericidal activity. Among these colistin components, two are predominant, colistin A and colistin B. Colistin A and B are concentration-dependent antibiotics, active against multidrug-resistant (MDR) Gram-negative bacteria (GNB) such as carbapenemase-producing *Enterobacterales, Pseudomonas aeruginosa* and *Acinetobacter baumannii*. CMS is an inexpensive antibiotic massively used worldwide and considered as an essential antimicrobial agent by the World Health Organization [2]. The CMS dose is labelled as “colistin base activity (CBA in mg)” in North and South America, Singapore, Malaysia, New Zealand and Australia and as “international units (IU)” in Europe and India. Milligrams of CBA and IU are expressions of the antibacterial activity measured in vitro and do not reflect a mass unit. The equivalence between absolute mass of CMS and IU or CBA is the following:
~ **80** *mg* CMS = **one***million IU* CMS = ~ **33.3** *mg* CBA.



This narrative review is focused on nebulized CMS as a treatment of ventilator-associated pneumonia (VAP) in critically ill patients. It has three aims: (1) to report the historical background supporting its use in critically ill patients; (2) to describe the complex and partially unknown PK/PD of intravenous and nebulized CMS; (3) to suggest future research priorities for CMS nebulization in patients with VAP caused by extensive drug-resistant (XDR) GNB.

## 2. Historical Background

Oral colistin was first used in veterinary medicine in 1952 and is still widely used in pigs to treat or prevent intestinal infections [3]. The worldwide prophylactic administration of colistin in swine industrial production is considered a major source of emerging colistin resistance [4].

### 2.1. Prophylaxis of Gram-Negative Bacteria Pneumonia

From the early 1970s to the mid-1990s, polymyxins were topically administered for prophylaxis of GNB pneumonia. In 744 spontaneously breathing or mechanically ventilated critically ill patients, polymyxin B was sprayed into the posterior pharynx and/or instilled in the endotracheal tube eight times a day [5,6]. This prospective double-blind study performed over an 11-month period, using alternating two-month cycles of placebo or polymyxin B, significantly reduced both bronchial GNB airway colonization and reduced the incidence of GNB nosocomial pneumonia without any apparent emergence of polymyxin resistance. In a subsequent series of 292 patients who received a daily intrapharyngeal and intratracheal prophylactic administration of polymyxin B during a six month period, the incidence of nosocomial pneumonia was significantly reduced, although pneumonias that did occur were more likely to be caused by polymyxin-resistant GNB [7]. Prolonged intratracheal prophylactic administration of polymyxins was subsequently considered as potentially dangerous and abandoned. Ten years later, nebulized colistin was advanced as a treatment for spontaneously breathing patients with cystic fibrosis and bronchial superinfection [8]. Over the next decade, colistin was administered intratracheally to prevent lung superinfection in baboons with oleic-acid- or hyperoxia-induced lung injury [9,10] and VAP in critically ill patients [11]. In the latter prospective before after study, the incidence of nosocomial GNB bronchopneumonia was significantly reduced in 347 critically ill patients on mechanical ventilation who received a two-week regimen of intratracheal CMS (eight endotracheal instillations per day of 200,000 IU). The incidence of nosocomial pneumonia was 40% in control patients (no prophylaxis by CMS) and 28% in the patients who received prophylactic CMS (*p* < 0.001). Over the two-year study period, there was no emergence of CMS resistance. Mortality was not significantly influenced by the prophylactic administration of CMS. These results were confirmed 20 years later in a single-center, two-arm, randomized, open-label, controlled trial performed in 186 critically ill patients on mechanical ventilation for > 48 h [12]. Eighty four patients received nebulized CMS (500,000 IU×3/24 h for 10 days) and 84 received nebulized saline during the same period. Nebulized CMS significantly reduced the incidence of MDR GNB VAP and improved the survival rate of patients with VAP and without evidence of increased resistance to colistin.

### 2.2. Treatment of MDR GNB Ventilator-Associated Pneumonia

In the early 2000s, intravenous [13,14] and nebulized [15] colistin became increasingly used to treat VAP and ventilator-associated tracheobronchitis (VAT) caused by MDR GNB. Over the first decade of the 2000s, the incidence of MDR GNB markedly increased worldwide and *P. aeruginosa and A. baumannii*, with extensive drug resistance (XDR) became the most common microorganisms causing VAP in Europe and Asia [16,17], and CMS often remained the only antimicrobial agent available against these pathogens.

Between 2005 and 2019, 25 articles reporting the use of nebulized CMS as a treatment of MDR VAP and VAT were published [15,18,19,20,21,22,23,24,25,26,27,28,29,30,31,32,33,34,35,36,37,38,39,40,41]. Nineteen were retrospective studies [15,18,19,20,21,23,24,25,26,27,29,30,31,32,33,35,36,38,39], four prospective observational studies [28,34,40,41] and two randomized control trials (RCTs) [22,37]. Twelve studies concerned the administration of nebulized CMS alone for treating VAP and VAT [15,18,19,22,26,27,28,32,33,34,36,41]. Thirteen studies compared the administration of nebulized CMS alone to the combination of intravenous and nebulized CMS for treating VAP [20,21,23,24,25,29,30,31,35,37,38,39,40]. Between 2005 and 2016, either low—1.2 to 4 million IU/day—[15,18,20,21,23,24,28,31,32,34] or high—4 million IU/day—[19,22,25,27,29,33,35,36] CMS doses were nebulized. From 2015, very high doses—9 to 15 million IU/day—were nebulized [37,38,39,40,41]. These very high doses were based on studies performed in 2008–2010 in anesthetized and mechanically ventilated piglets whose lungs were infected by the bronchial inoculation with high concentrations of P. aeruginosa [42]. The nebulization of 100,000 IU/kg of CMS could eradicate the causative microorganism from 67% of the infected pulmonary segments within 24 h. These very high doses are actually recommended [43]. In 2018, a meta-analysis performed on 12 studies published between 2005 and 2016 [15,18,19,22,26,27,28,32,33,34,36,37] reported the effectiveness of nebulized CMS as a monotherapy for treating respiratory tract infections caused by MDR and/or CMS only susceptible GNB [44]. As shown in Figure 1, the clinical and microbiological success rate was 70%, an efficacy similar to that observed in VAP caused by susceptible GNB treated by intravenous antibiotics. Three methodological concerns undermined the clinical relevance of the meta-analysis: concomitant administration of intravenous antibiotics active against the causative microorganism in seven studies [22,27,28,32,33,34,36,37], nebulization of low CMS doses in three studies [32,33,34] and optimization of nebulization technique in only two studies [26,28]. Despite these limitations, the meta-analysis brought strong evidence that nebulization of CMS could provide a clinical, radiological and microbiological cure of VAP caused by MDR GNB and, specifically, XDR *A. baumannii* (Figure 2).

In 2015, two meta-analyses [45,46], including ten studies performed between 2005 and 2014 [20,21,22,23,24,25,29,30,31,35], suggested that the association of nebulized and IV CMS as a treatment of VAP and VAT caused by MDR GNB was associated with better clinical and microbiological response and lower infection-related mortality than intravenous therapy alone. However, a further meta-analysis published in 2018 [47] concerning thirteen studies performed between 2005 and 2016 [20,21,23,24,25,29,30,31,35,48,49,50,51] did not confirm these benefits. The retrospective nature of studies, their heterogeneous protocols, the lack of optimization of the technique of nebulization and the variability of dosing restrict the validity of these different meta-analyses. There is clinical and experimental evidence that nebulization of high-dose CMS may be an efficient treatment of MDR GNB VAP and VAT. It is unclear whether nebulized high-dose CMS is equivalent or superior to the treatment by intravenous CMS or new cephalosporin/ß-lactamase inhibitors [52,53].

## 3. Structure–Activity Relationship

### 3.1. Chemical Structure and Antimicrobial Activity

As shown in Figure 3a, both molecules differ only by the radical occupying position 6 of the *N^α^* fatty acyl chain. PMB and colistin are composed of a mixture of active components differing by the type of fatty acyl chain linked to the N-terminal Dab residue (PMB 1-6 and colistin A-F). The *N*-terminal fatty acyl chain is crucial for the antimicrobial activity of polymyxins: it allows the disruption of the lipid A fatty acyl chains of the lipopolysaccharides (LPS) component of the outer membrane of GNB [54]. The polar residue of the heptapeptide also plays an important role for antimicrobial activity and LPS binding affinity. Of particular importance is the specific order of the cationic diaminobutyric acid (Dab) residues within the primary sequence that confers the proper spatial distribution of the positive charges for electrostatic interactions with the anionic phosphates of lipid A of the LPS. Last but not least, the unique three-dimensional architecture of PMB and colistin is required for both LPS binding and antimicrobial activity.

### 3.2. Mechanisms of Bacterial Killing

Polymyxins rapidly kill bacteria by disrupting the outer and inner membranes of GNB [55]. As there is only one amino acid difference between colistin and polymyxin B, it is generally considered that both antibiotics share the same mechanisms of action. As shown in Figure 4a and b, the initial adherence of colistin to the outer membrane occurs via electrostatic interactions between the Dab residues of the antibiotic and anionic phosphate of the lipid A moiety of LPS. The positively charged amine groups and the hydrophobic fatty acyl chains of colistin play important roles in the interaction with LPS. Physiologically, divalent cations Ca^++^ or Mg^++^ associated with lipid A phosphodiesters serve as bridges between adjacent LPS molecules and stabilize the outer membrane (Figure 4a). As cations Dab of colistin have an affinity for anions phosphate of Lipid A that is at least three times higher than the one of divalent cations [56], they competitively displace Ca^++^ or Mg^++^, disrupt the LPS bridges of the outer membrane and permit the colistin penetration and the leakage of cell content (Figure 4b). Colistin also acts through several other mechanisms [55]: oxidative stress death pathway via the production of reactive oxygen species (hydroxyl, superoxide and hydrogen peroxide radicals); vesicle–vesicle contact pathway where colistin, after transiting the outer membrane, induces the fusion of the inner leaflet of the outer membrane with the outer leaflet of the cytoplasmic membrane, leading to loss of phospholipids and cell death; inhibition of respiratory enzymes of the tricarboxylic acid cycle. 

### 3.3. Anti-Endotoxin Activity

Colistin exerts a potent anti-endotoxin effect by inhibiting the activity of lipid A. Colistin binds to the negatively charged lipid A region of the lipopolysaccharide via electrostatic and hydrophobic interactions at a ratio of one colistin molecule to one LPS monomer unit [57]. As shown in Figure 4c, colistin aggregates LPS released in high concentrations following the breakdown of the bacterial membrane and, thereby, decreases the endotoxin’s ability to induce shock through the release of pro-inflammatory cytokines (tumor necrosis factor-alpha, interleukins and monocyte chemoattractant protein 1).

The anti-endotoxin activity of polymyxins has been considered as a potential treatment of sepsis [58]. Hemoperfusion using a polymyxin B-immobilized fiber blood purification column was developed in Japan in the nineties and proposed to remove circulating endotoxin in sepsis and shock [59]. Between 2004 and 2016, three RCTs were performed in patients with intra-abdominal infection and VAP, testing the effect of two sessions of polymyxin B hemoperfusion on mortality and severity of organ failure [60,61,62]. Unfortunately, polymyxin B hemoperfusion failed to reduce mortality rate and severity of organ failure, particularly in patients with high circulating endotoxin levels. Although a recent meta-analysis suggested a benefit of polymyxin hemoperfusion in patients with less severe forms of septic shock [59], this technique cannot be recommended as a routine treatment of severe sepsis.

### 3.4. Mechanisms of Resistance

GNB resistance to colistin can be chromosomally encoded, resulting from genetic mutations or can be plasmid-mediated, raising concern for potential dissemination. The main mechanism of resistance is a reduction in the negative charges of the LPS that physiologically allow the electrostatic interaction of colistin with the outer membrane. Replacing the anionic phosphate groups of lipid A by cationic moieties hinders the binding and preclude the bactericidal activity of colistin [55]. Genetic mutations modify the structure of the bacterial membrane in several ways [63]: addition of capsular polysaccharides or cationic moieties to the LPS hiding the colistin binding sites; loss of the LPS; porin modifications with overexpression of efflux pump systems; enzymatic inactivation of colistin. Heteroresistance resulting from bacterial exposure to suboptimal colistin dosages represents a potential source of colistin resistance and should be considered as an emerging menace [55].

The horizontal transfer of plasmid-borne genes of the family *mobile colistin resistance* (*mcr*) is another mechanism of colistin resistance. *Mcr* genes encode a phosphoethanolamine transferase leading to the addition of a phosphoethanolamine moiety to the lipid A of LPS, increasing the cationic charges on LPS and, consequently, limiting the binding of colistin to LPS. Up to now, nine *mcr* alleles have been reported, *mcr-1* to *mcr-9* [55]. As shown in Figure 5, *mcr-1* was reported on all continents after being initially identified from chickens in China three decades ago when colistin started to be used in food-producing animals [63]. The main reason for dissemination of colistin resistance worldwide is the large and indiscriminate use of polymyxins in veterinary medicine [3]. In pigs and calves, oral colistin is administered as a prophylaxis of gastrointestinal infections caused by *Enterobacterales.* In swine industrial production, colistin is administered by feed and drinking water in the entire farm, involving indiscriminately both healthy animals and animals with clinical symptoms. In 2016, the European Medicines Agency recommended to ban colistin therapies for prophylactic purposes, as it carries a high risk of the emergence of resistance. Despite this recommendation, prophylactic administration of colistin is still a common practice worldwide, especially in Asia, to support farm animal production. Interestingly, in the Netherlands, colistin is routinely used in selective digestive decontamination regimens without a report of alarming rates of resistance.

## 4. Pharmacokinetics

### 4.1. Pharmacokinetics of Intravenous Colistimethate Sodium

CMS is a complex mixture of up to ~30 methanesulfonated derivatives produced by the reaction of colistin with formaldehyde and sodium bisulfite. The composition of CMS pharmaceutical products may vary from brand-to-brand and even from batch-to-batch. CMS acts as a polyanionic inactive prodrug that is less nephrotoxic than colistin sulfate. As shown in Figure 3b, methanesulfonate moieties are masking the primary amines of the Dab residues, are negatively charged at physiological pH and preclude the interaction of CMS with anionic phospholipids of LPS. As a consequence, CMS lacks any antibacterial activity [64]. 

As shown in Figure 6a, CMS is rapidly and massively cleared from the blood by glomerular filtration and tubular excretion. Additionally CMS is spontaneously converted to colistin by hydrolysis, a necessary step to achieve antibacterial activity. The renal elimination of CMS is quantitatively greater than its spontaneous hydrolysis into colistin. In patients with normal renal function, approximately 20–25% of CMS is converted to colistin. The various composition in methanesulfonated derivatives of the different pharmaceutical products affects the spontaneous hydrolysis of CMS in biological fluids and is associated with a brand-to-brand and batch-to-batch interindividual variability in the rate of conversion. Adding to the complexity of pharmacokinetics, colistin resulting from the spontaneous hydrolysis of plasma CMS is filtered by the kidney and largely reabsorbed by the renal tubules. Although only a minor fraction is directly excreted in urine, urinary concentrations of colistin can be high, resulting from the spontaneous hydrolysis of CMS within renal tubules and bladder [64]. The mechanism of elimination of polymyxins is far from being elucidated [65]. Biliary excretion is likely the predominant pathway, as the different components of polymyxin B have been detected in bile [66,67,68].

Because renal elimination of CMS is much more rapid than its spontaneous conversion to colistin, it is necessary to administer about 4–5 times the amount of CMS to generate colistin plasma concentrations above the minimum inhibitory concentrations required for a bactericidal activity. As shown in Figure 6d–f, the disposition of CMS is best described by a two compartment linear model, whereas the disposition of colistin is best described by a one compartment model [69,70]. Following a single administration per day, the colistin profile is flatter than the CMS profile, offering the possibility of a longer half-life and a prolonged antibacterial effect if intravenous administrations are repeated 2-3 times a day. As in vivo conversion of CMS to colistin is slow, incomplete and variable, achievement of colistin plasma concentrations ≥ minimum inhibitory concentrations can be facilitated by a loading dose of 9 million IU. However, even with a loading dose and high daily doses (up to three times per day of 3 million IU), the brand-to-brand and batch-to-batch variability in CMS components may decrease the rate of conversion to colistin, although the CMS renal clearance remains rapid and efficient [71]. As a consequence, if renal function is normal, an increase in inter-individual pharmacokinetics variability may result, precluding the achievement of optimal colistin plasma concentrations required to obtain an efficient bactericidal effect. This may have a negative impact on prognosis given the link between delayed initiation of appropriate antibiotic therapy and patient outcome. Thus, the intravenous administration of CMS, a complex prodrug whose conversion to colistin, the active antibiotic, is far slower than its renal clearance and does not create optimal conditions to treat efficiently VAP caused by MDR GNB. As a consequence, CMS selected by physicians ranged between from 2.3 and 12 million IU/day, and plasma concentrations were measured using high-performance liquid chromatography. (d–f) Representative individual population pharmacokinetic model fits of CMS and colistin. Panel d illustrates a critically ill patient not on renal replacement. Panels e and f are representative of a patient on hemodialysis or on continuous renal; if intravenous CMS is used, concentrations should be measured regularly to detect suboptimal dosage. Such a difficult issue can be partially resolved by nebulization of CMS: the prodrug trapped in the distal lung undergoes progressive local conversion to colistin with a low systemic absorption and limited renal elimination.

### 4.2. Pharmacokinetics of Nebulized Colistimethate Sodium

Pharmacokinetics of nebulized CMS are incompletely understood mainly because of the difficulty to assessing lung interstitial concentrations in human studies. Indeed, the contamination of the bronchoscope by bronchial secretions during the bronchoalveolar procedure skews the interpretation of epithelial lining fluid (ELF) concentrations following antibiotic nebulization [53,72] and leads to a gross overestimation of interstitial space fluid concentrations [73,74]. Animal studies demonstrate that high colistin concentrations measured in post mortem subpleural lung specimens are high following nebulization of CMS 100,000/kg × 2/24 h in ventilated piglets with massive *Pseudomonas aeruginosa* inoculation pneumonia [42]. Lung homogenate colistin concentrations depend on both the severity of aeration loss and histologic grade (Figure 8a,b) with lower colistin lung concentration associated with a more severe histological grade of pneumonia. Colistin concentrations > five times minimum inhibitory concentrations are exclusively obtained in lung regions with a moderate severity of pneumonia and relative preservation of lung aeration, whereas in pulmonary segments with confluent pneumonia, colistin concentrations are in the range of minimum inhibitory concentrations. These high lung tissue concentrations, which underestimate interstitial space fluid concentrations due to the dilution effect of pulmonary cells and vessels, are associated with a rapid and potent bacterial killing [42].

Nebulization of high doses of CMS results in high lung tissue colistin concentrations with correspondingly low plasma colistin concentrations (< 2 µg/mL) (Figure 6c) suggesting limited diffusion into the systemic compartment [26,28,41,42,75,76,77,78,79,80,81,82]. After the initial CMS nebulization of 2 million IU (Figure 8a,b), CMS and colistin plasma concentrations show quite similar pharmacokinetic profiles [79]: an early peak concentration for CMS (30 min), a delayed peak concentration for colistin (3 h) and a slow and progressive decrease in concentrations over the following hours. After 2–3 days of CMS nebulization at a dose of 4 million IU three times a day (Figure 7c,d), similar pharmacokinetic profiles were observed [81]. Repeated nebulized CMS doses result in increased CMS plasma concentrations. In the majority of patients, there was a slight increase in colistin plasma concentration when the nebulized CMS dose was increased from 0.5 to 4 million IU (Figure 7b,d). In a few patients, however, colistin plasma concentrations markedly increase, approaching concentrations observed after intravenous administration and plateauing over time (Figure 7d). These results suggest that CMS and colistin accumulate in the lung compartment. Colistin plasma concentrations remain low (< 2 µg/mL) in the majority of patients due to the slow diffuse of both CMS and colistin into the systemic circulation, rapid renal elimination of CMS and slow hydrolysis of CMS to colistin.

## 5. Pharmacodynamics

### 5.1. Concentration-Dependant Effect of Colistin and Post-Antibiotic Effect

As shown in Figure 8c, colistin shows rapid concentration-dependent killing against GNB at clinically achievable concentrations [83]. Re-growth often occurs as early as within 2 h of the initial exposure. An inoculum effect (the bactericidal effect of a given colistin dose decreases at high inoculum) has been reported in vitro [84]. Using neutropenic mouse lung infection models, the ratio of the area under the free concentration–time curve to the MIC (*f*AUC/MIC) best describes the antimicrobial activity of colistin [83]. For MDR strains, such as *P. aeruginosa* and *A. baumannii*, an *f*AUC/MIC value of 7.4–13.7 and 7.4–17.6 is required for a 2 log10 reduction in bacterial load. Following intravenous administration, 2 log10 killing in the lungs cannot be achieved with doses as high as 50,000/kg, likely due to limited drug exposure in the lungs. Experimental and clinical PK/PD data clearly indicate that intravenous colistin (and CMS) has limited efficacy against respiratory tract infections [85]. Pharmacodynamics of nebulized CMS have not been described. Experimental studies using intrapulmonary microdialysis are required to assess interstitial lung CMS and colistin concentrations changes over time and their effect on bacterial killing [86].

### 5.2. Toxicity and Toxicodynamics of Intravenous Colistin

Neuromuscular toxicity, nephrotoxicity and bronchoconstriction are most common adverse events associated with CMS administration. Polymyxin-induced neuropathy and myopathy is rarely seen [87]. Nephrotoxicity is the most common side effect observed both with colistin and PMB and most commonly results from the intravenous administration of CMS [88]. Patients with high creatinine clearance ≥ 80 mL/min are most likely to develop nephrotoxicity [89] due to enhanced renal elimination of CMS (Figure 6a) and, paradoxically, have low plasma colistin concentrations [81,90]. Nephrotoxicity can be detected two days after initiation of intravenous CMS, with the majority of cases occurring after 15 days of therapy. Commonly, colistin-induced nephrotoxicity is reversible. As plasma colistin concentrations remain < 2 μg/mL following the nebulization of high doses of CMS (Figure 6c–f and Figure 6b,d), the risk of nephrotoxicity of nebulized CMS is low.

Cell culture and animal studies demonstrate that colistin accumulates in renal tubular cells. Urinary colistin is reabsorbed via active uptake mechanisms mediated by megalin and oligopetide transporter 2 [88]. The resultant high intratubular colistin concentrations causes mitochondrial damage, loss of cytoplasmic membrane potential, apoptosis and cell cycle arrest [91,92]. Immunostaining studies performed in rodents have shown predominant accumulation of polymyxins in proximal tubular cells of the renal cortex. Detailed mechanisms of the uptake by renal tubular cells and subsequent cell death remain to be elucidated. Specifically, there is a paucity of information on the relationships between chemical structure and nephrotoxicity [54]. The concomitant administration of the antioxidant ascorbic acid has provided contradictory findings [93,94], necessitating further examination.

A meta-analysis was performed on 12 studies including 373 patients treated by nebulized CMS monotherapy for VAP and ventilator-associated tracheobronchitis. Acute kidney injury was observed in 20% of treated patients compared to 31% in control patients. Neuromuscular toxicity was observed in 3% of patients and bronchospasm in 2% of patients [44].

## 6. Administration and Dosing of Nebulized Colistimethate Sodium

### 6.1. Technique of Nebulization

Optimizing lung deposition of CMS to provide effective bacterial killing in critically ill patients with VAP and VAT requires a specific nebulization strategy including [43,53,94]: (1) the preferential use of vibrating mesh nebulizers positioned 15 cm before the Y piece; (2) the use of continuous rather than breath-synchronous nebulization to allow the nebulization of high doses of CMS; (3) the use of specifically designed respiratory circuits with smooth inner surfaces and avoiding sharp angles to decrease turbulence and circuit deposition; (4) the use of specific ventilator settings to limit circuit and tracheobronchial impaction of aerosolized particles: volume controlled mode with constant inspiratory flow, tidal volume 8 mL/kg, respiratory frequency 12 to 15 bpm, inspiratory: expiratory ratio 50%, inspiratory pause 20% and positive end-expiratory pressure 5 to 10 cm H_2_O; (5) the administration of a short-acting sedative agent to ensure coordination between the patient and the ventilator; (6) the insertion of a filter on the expiratory limb to protect the ventilator flow device. This filter should be changed between each nebulization to avoid expiratory flow obstruction; (7) the removal of heat and moisture exchanger and the interruption of the conventional heated humidifier to avoid massive trapping and condensation of aerosolized particles.

### 6.2. Nebulized Doses

Owing to the PKPD characteristics of CMS and its excellent bronchial tolerability [44], nebulization doses as high as 15 million IU/24h (200,000 IU/kg) can be recommended [43,53]. If the technique of nebulization is optimized, very high colistin lung tissue concentrations are obtained. The infected lung acts as a CMS reservoir where spontaneous hydrolysis into colistin occurs slowly, providing delayed but efficient continuous bacterial killing. In critically ill patients with VAP and VAT, the bronchial inoculum is high, ≥ 10^6^ colony forming units per mL. The inoculum effect of CMS [84] and the slow process of intrapulmonary colistin formation incite to the nebulization of high dose. Intrapulmonary CMS partly diffuses into the systemic compartment, and 17% of the nebulized dose is rapidly eliminated by the kidney [81]. The slow hydrolysis of CMS into colistin in the plasma and urine keeps colistin plasma concentrations low whatever the CMS nebulization dosing. To limit each nebulization time to ≤ 60 min, three nebulizations of 5 million IU can be administered/24 h. 

### 6.3. Conditions of Administration

CMS manufacturers recommend dissolving each 1 million IU of CMS with 3 mL of normal saline solution. Therefore, 15 mL is required to nebulize a dose of 5 million IU. As the inner volume of most nebulizer chambers range between 6 and 10 mL, the nebulizer needs to be filled at least twice, which lengthens the nebulization time beyond 60 min, increases nurses’ workload and carries the risk of incomplete administration. It has been shown that reducing the diluent volume to 6 mL for nebulizing 4 million IU improves colistin stability and does not modify aerosol characteristics nor plasma and urine PKPD [81]. Therefore, diluting 5 million IU of CMS powder with 6 mL of normal saline is possible. 

CMS and colistin are not stable in various aqueous media [81,95]. It is highly recommended to nebulize solutions of CMS that are reconstituted just before use. A 29-year-old woman with cystic fibrosis superinfected by *Pseudomonas aeruginosa* died from an acute respiratory distress syndrome after receiving a CMS nebulization of a pharmacy-compounded premixed solution that was 5 weeks old (with the stated expiration date not yet reached). Toxic degradation products resulting from the long conservation of the aqueous solution were considered as responsible for the fatal outcome [96].

## 7. Future Research

### 7.1. Concerns on the Use of Intravenous Colistimethate Sodium

Following intravenous administration, 70% of a CMS dose is rapidly cleared by the kidney, whereas 20–25% is slowly hydrolyzed to colistin through a process that takes more than 36 h. Penetration into all organs except the kidneys and biliary tract is also quite limited. PKPD studies indicate that it is difficult to reach bactericidal colistin concentrations at the site of infection in patients with normal renal function without exceeding the recommended plasma concentrations of 2 mg/L, above which the risk of nephrotoxicity markedly increases [52]. Microdialysis studies performed in anaesthetized healthy female pigs have shown that physico-chemical properties play a pivotal role for antibiotic penetration across the vascular endothelium, basement membrane and the respiratory epithelium. High lipophilicity, low molecular mass, polarity and charge at physiological pH favor penetration into pulmonary epithelial lining fluid. As CMS and colistin have one of the highest molecular masses and polarities, are significantly charged (Figure 3a) and highly hydrophilic, their ability to cross the alveolo-capillary membrane is among the lowest [86]. Owing to rapid renal clearance and limited pulmonary diffusion and despite optimized CMS dosing, intravenous colistin does not appear optimal to treat lower respiratory tract infections [52,97].

### 7.2. Concerns on the Use of Nebulized Colistimethate Sodium

As systemic diffusion of nebulized CMS is limited and its hydrolysis in colistin is slow, plasma concentrations of colistin following CMS nebulization remain lower than minimal inhibitory concentrations in the majority of patients (Figure 8b,d). Therefore, nebulized colistin alone cannot be considered as a safe therapeutic option in patients with bacteremic VAP. Combination of nebulized CMS with intravenous new cephalosporines/ß-lactamase inhibitors or intravenous CMS should be considered.

Another issue concerns the diffusion of nebulized CMS into consolidated lung areas characterizing confluent and lobar VAP. In theory, the lack of lung aeration should preclude the penetration of nebulized CMS into the consolidated infected lung parenchyma. Interestingly, experimental studies have repeatedly shown that, although lung tissue concentrations decrease with the aeration loss, they remain largely above minimal inhibitory concentrations in the majority of animals with consolidated infected lung regions [42,98,99,100]. As shown in Figure 9, these findings were observed in ventilated and anesthetized piglets treated by: (1) high dose of nebulized amikacin for an inoculation pneumonia caused by sensitive *Escherichia coli* [98,99]; (2) high dose of nebulized CMS for an inoculation pneumonia caused by sensitive *Pseudomonas aeruginosa* [42]; (3) high dose of nebulized ceftazidime for an inoculation pneumonia caused by partially resistant *Pseudomonas aeruginosa* [98]. After the nebulization of CMS at a dose of 130,000 IU /kg per day, lung tissue concentrations ranged between 0.8 and 7 µg∙g^−1^ in consolidated lung areas. Among 13 lung segments with consolidation, nine had tissue lung concentrations ≥ 2 µg∙g^−1^, the MIC of the inoculated Pseudomonas aeruginosa strain. As lung homogenate tissue concentrations markedly underestimate interstitial space fluid concentrations, it can be hypothesized that true concentrations at the site of infection were higher. As for many nebulized antibiotics, colistin tissue concentrations decrease with the degree of aeration loss [42] but remain greater than MIC in the majority of consolidated lung areas (zero aeration). Moreover, a bactericidal effect is observed in the majority of lung segments with severe pneumonia, confirming the existence of bactericidal concentrations at the site of infection [42]. The exact mechanisms by which nebulized colistin diffuse into the consolidated lung parenchyma remain speculative and likely multifactorial: bronchiolar distension and pseudocysts are possible ways of penetration of nebulized antibiotics into consolidated lung regions [101], as well as diffusion of nebulized antibiotics through the bronchial wall of non-obstructed distal bronchioles penetrating within consolidated pulmonary segments. By increasing nebulized dose of CMS above 130,000 IU/kg, it can be reasonably expected that bactericidal colistin concentrations can be reached in lobar pneumonia. This forms the rationale for using nebulized CMS doses as high as 200,000 IU/kg in patients with XDR VAP (the equivalent of 5 million IU at 3 h interval in a 75 kg adult patient). 

Last but not least, it is difficult to assess interstitial space fluid concentrations from epithelial lining fluid concentrations measured from a bronchoalveolar lavage sample [43,72]. In a porcin model of inoculation pneumonia caused by multiresistant *Pseudomonas aeruginosa* and treated by high-dose nebulized amikacin and/or Fosfomycin combined or not with intravenous meropenem, tracheal antibiotic concentrations were compared to lung tissue [102]. Due to the bronchial deposition of aerosolized particles, peak tracheal amikacin concentrations were 500 higher than lung tissue concentrations. Interestingly, the nebulization of high-dose amikacin did not decrease bacterial concentrations in the infected lung parenchyma but eradicated the inoculated Pseudomonas aeruginosa from upper airways, although strains were resistant to amikacin (MIC > 32 µg∙mL^−1^) [102]. These data clearly suggest that the nebulization of high-dose amikacin or CMS are efficient to treat VAT caused by GNB resistant to both antibiotics.

### 7.3. Substitution Rather Than Adjunctive Colistimethate Sodium Therapy

As the treatment efficiency of intravenous CMS is frequently suboptimal, adjunctive therapy (nebulization + intravenous CMS) appears unwarranted in non-bacteremic VAP. Compared to intravenous CMS alone, adjunctive therapy increases lung tissue concentrations without reducing plasma concentrations. Therefore, it likely improves efficacy without decreasing toxicity risk. Compared to adjunctive therapy, substitution therapy (nebulized CMS alone) markedly reduces colistin plasma concentrations and decreases the risk of toxicity, as shown in a recent meta-analysis [103]. Therefore, the European Society of Clinical Microbiology and Infectious Diseases recommended to compare substitution therapy rather than adjunctive therapy with intravenous administration in future randomized controlled trials (RCTs) [104]. This recommendation is not valid for patients with bacteremic VAP where a combination of nebulized and intravenous CMS should be used.

### 7.4. Experimental Studies Using Intrapulmonary Microdialysis Are Required

Assessing intrapulmonary PKPD of nebulized CMS and colistin is quasi impossible in patients with VAP, because the bronchoalveolar samples serving for measuring epithelial lining fluid concentrations are contaminated by nebulized CMS and colistin [43,53,73,75,76]. For the same reason, bacteriological cure cannot be reliably assessed by quantitative culture of bronchoalveolar lavage fluid in patients receiving nebulized antibiotics. Only measurements of interstitial space fluid concentrations by intrapulmonary microdialysis and quantitative culture of postmortem lung biopsies can provide an accurate view of PKPD and bactericidal activity of CMS and colistin.

Future experimental studies using intrapulmonary microdialysis should describe the time-dependent profile of CMS and colistin interstitial space fluid concentrations after the nebulization of the first and the following nebulizations of 200,000 IU/kg. In parallel, systemic, biliary and urinary CMS and colistin concentrations should be assessed. Another experiment should verify that increasing concentrations of nebulized CMS (from 50,000 to 200,000 IU/kg) are associated with increasing interstitial space fluid concentrations of colistin. Again, systemic, biliary and urinary CMS and colistin concentrations should be assessed in parallel.

### 7.5. Future Randomized Multicenter Controlled Trials

A number of new antimicrobial agents are now available with activity against MDR GNB. Ceftazidime avibactam and ceftolozane tazobactam are active against MDR-XDR Enterobacterales (a key challenge in countries with middle and low outcome). Ceftazidime avibactam and ceftolozane tazobactam are active against MDR-XDR Enterobacterales. Ceftazidime avibactam is active against *Klebsiella pneumoniae* producing carbapenemase and oxacillinase and ceftolozane tazobactam against *Pseudomonas aeruginosa* with non-enzyme mediated carbapenem resistance. Cefiderocol or eravacycline are active against carbapenem-resistant *Acinetobacter baumannii*.

CMS remains a key agent in the treatment of VAP and VAT caused by MDR GNB, but the optimal regimen is currently unclear. Randomized controlled trials are urgently required to identify optimal treatment strategies for patients with VAP and VAT caused by MDR GNB. Superiority multicenter RCTs comparing nebulized CMS alone with the intravenous administration of CMS are required to guide treatment when new cephalosporines/ß-lactamase inhibitors are not available or when VAP or VAT is caused by XDR GNB resistant to the new cephalosporines/ß-lactamase inhibitors. Where new cephalosporines/ß-lactamase inhibitors are available, non-inferiority multicenter RCTs should compare the nebulization of CMS alone with the parenteral administration of new cephalosporines/ß-lactamase inhibitors. Expected benefits from nebulized CMS are a more rapid clinical cure, a reduction in the duration of mechanical ventilation, less nephrotoxicity and a shorter duration of antibiotic administration. As lung deposition of nebulized CMS decreases with lung aeration [42], patients should be included as early as possible in the different multicenter RCTs. Bacteriological cure should be determined at least 24 h following the last CMS nebulization to avoid the presence of nebulized colistin in the sample that could artifactually prevent bacterial growth and provide a false negative result.

## 8. Conclusions

Although well-designed multicenter RCTs are lacking, there is a body of evidence suggesting that nebulized CMS is efficient for treating lower respiratory tract infections caused by MDR GNB. The complex PKPD of the intravenous prodrug CMS, characterized by a rapid renal elimination and a slow hydrolysis in active colistin, advocates for nebulization rather than intravenous administration. Experimental studies using intrapulmonary microdialysis are urgently needed to characterize lung PKPD and optimize CMS nebulization, but preliminary studies suggested nebulization of CMS results in high intrapulmonary concentration of colistin. Future superiority multicenter RCTs should compare high doses of nebulized CMS to high doses of intravenous CMS in VAP and VAT caused by extensive drug-resistant GNB not sensitive to aminoglycosides and/or new cephalosporines/ß-lactamase inhibitors. Noninferiority multicenter RCTs should compare high doses of nebulized CMS alone to new cephalosporines/ß-lactamase inhibitors in VAP and VAT caused by extensive drug-resistant GNB.

## Figures and Tables

**Figure 1 microorganisms-09-01154-f001:**
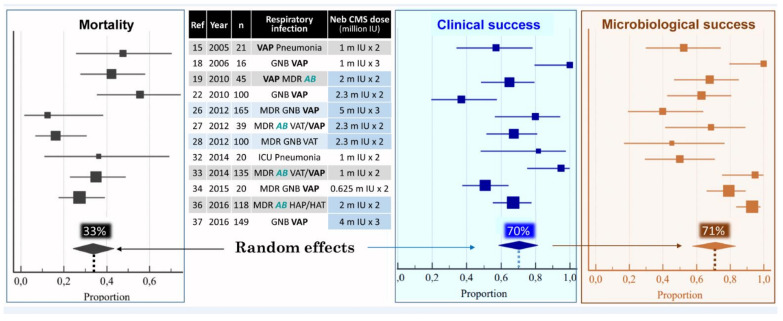
Pooled analysis of mortality, clinical and microbiological success among 908 patients treated with nebulized colistimethate sodium (CMS) for ventilator-associated pneumonia (VAP) or ventilator-associated tracheobronchitis (VAT) caused by multidrug-resistant (MDR) Gram-negative bacteria (GNB), particularly *Acinetobacter baumannii* (AB). Squares = proportion in each study; horizontal lines = 95% CI; diamonds = pooled proportion for the 12 studies. Doses of nebulized CMS are expressed in million international unit (IU). Adapted with permission from ref. [44]. Copyright 2017 Elsevier.

**Figure 2 microorganisms-09-01154-f002:**
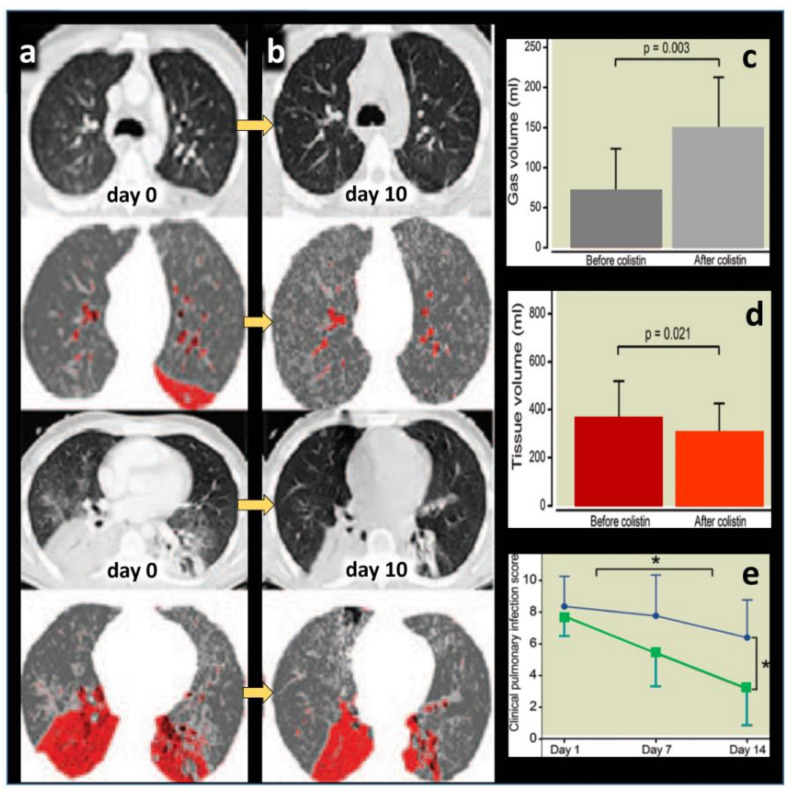
Representative computed tomography images obtained in a patient with ventilator-associated pneumonia (VAP) caused by multidrug-resistant *Pseudomonas aeruginosa* and treated by nebulized colistimethate sodium (CMS) 5 million international units × 3 /24 h for 10 days. A color encoding system identifies normally aerated lung regions (dark grey), poorly aerated lung regions (light grey) and nonaerated consolidated lung regions (red). (**a**) Contiguous 10 mm thick computed tomography sections obtained before nebulization (day 0) shows bilateral consolidation of lower lobes with disseminated foci of interstitial pneumonia in upper lobes. (**b**) Ten days later, lung consolidations are partially reaerated, attesting to the clinical efficiency of nebulized CMS monotherapy. (**c**,**d**) Computed tomography quantitative assessment of gas volume (aeration) and tissue volume (inflammation/infection) before and after CMS (colistin) nebulization in seven patients with VAP caused by MDR *Pseudomonas aeruginosa.* Nebulized CMS monotherapy was associated with a significant re-aeration and decrease in inflammation/infection. (**e**) Changes in Clinical Pulmonary Infection Score in 29 patients with VAP caused by MDR *Pseudomonas aeruginosa* or *Acinetobacter baumannii* successfully treated by nebulized CMS (green color) and in 13 patients with VAP caused by MDR *Pseudomonas aeruginosa* or *Acinetobacter baumannii* unsuccessfully treated by nebulized CMS (blue color). * indicates *p* < 0.001 [26].

**Figure 3 microorganisms-09-01154-f003:**
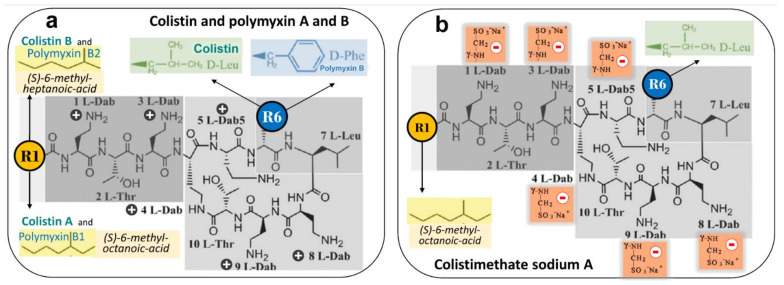
Chemical structures of polymyxin B (PMB), colistin and colistimethate sodium. (**a**) PMB is characterized by D-Phenyl and colistin by D-Leuc in position 6. Each antibiotic is a mixture of active components differing by the type of fatty acyl chain linked to the N-terminal diaminobutyric acid (Dab) residue. At physiological pH, Dab are positively charged + and interact with anionic phosphates of lipid A of LPS, thereby disrupting the bacterial outer membrane; colistin A and PMB1 = (S)-6-methyl-octanoic acid; colistin B and PMB2 = (S)-6-methyl-heptanoic acid; colistin C and PMB3 = octanoyl acid; colistin D and PMB4 = heptanoyl acid; colistin E and PMB5 = nonanoyl; colistin F and PMB6 = 3-hydroxy-6- methyloctanoyl acid. Light grey identifies the polar residues of the heptapeptide, light purple the hydrophobic motif within the heptapeptide ring and dark grey the *N*-terminal fatty acid analogues. (**b**) Colistimethate sodium, the inactive prodrug of colistin, is prepared from colistin by reaction of the free γ-amino groups of the Dab residues with formaldehyde followed by sodium bisulfite (methanesulfonate moieties). Colistimethate sodium A and B are defined by the fatty acid chain linked to Dab in position 1: (S)-6-methyl octanoic acid for colistimethate A and (S)-6-methyl heptanoic acid for colistimethate B. At physiological pH, methanesulfonate moieties are negatively charged—and Dab cannot interact anymore with anionic phosphates of lipid A of LPS, thereby precluding any bactericidal effect (see Figure 4). Adapted with permission from ref. [54]. Copyright 2009 American Chemical Society.

**Figure 4 microorganisms-09-01154-f004:**
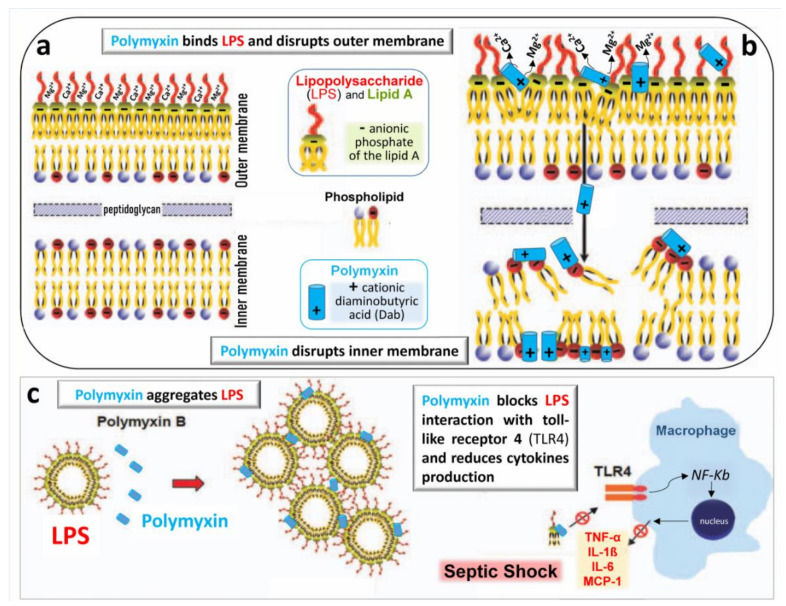
Schematic representation of the bactericidal action of polymyxins. (**a**) The outer membrane of the Gram-negative bacterial wall is stabilized by the electrostatic interaction between divalent cations Ca^++^ and Mg^++^ and negatively charged phosphodiesters of lipid A of lipopolysaccharide (LPS), thereby creating a permeability barrier against harmful external agents. (**b**) Polymyxins disrupt the physiological bridges LPS/lipid A. The positively charged Dab interacts electrostatically with negatively charged phospholipids of the LPS, producing the leakage of cellular components through the disrupted bacterial membrane. The mechanisms by which polymyxins disrupt the bacterial inner membrane remain undetermined. (**c**) Polymyxins aggregate free LPS released from the bacterial wall and block LPS interaction with macrophage receptor, TLR4. The NF-kB pathway is no longer stimulated, and release of pro-inflammatory cytokines such as tumor-necrosing factor-α (TNF-α), interleukins 1β and 6 (IL-1β and IL6) and monocyte chemoattractant protein 1 (MCP-1) is limited, reducing the sepsis severity [26].

**Figure 5 microorganisms-09-01154-f005:**
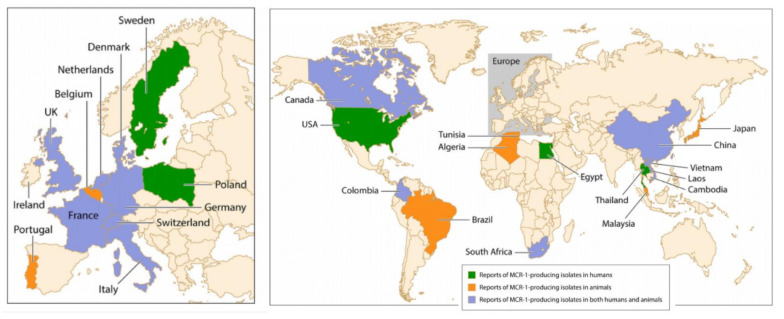
Worldwide distribution of *mcr-1*-producing isolates in humans and animals [63].

**Figure 6 microorganisms-09-01154-f006:**
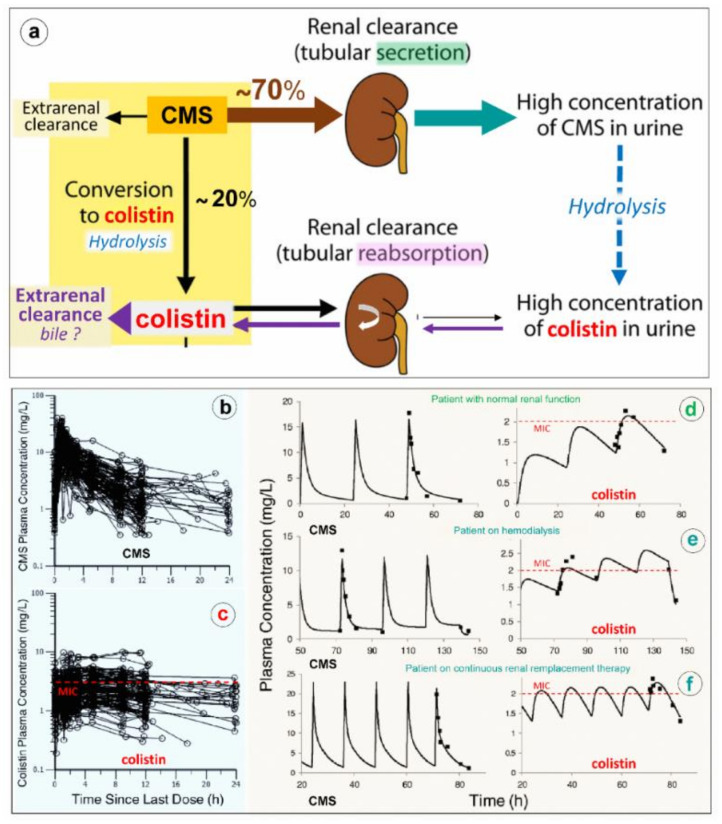
Elimination and pharmacokinetics of colistimethate sodium (CMS) and colistin. (**a**) Schematic representation of elimination pathways for CMS and colistin. Arrow thickness indicates the relative magnitude of each pathway when kidney function is normal. CMS includes all partially methanesulfonated derivatives of colistin. After intravenous administration of CMS, extensive renal excretion occurs, with some of the excreted CMS being converted to colistin within the urinary tract. Colistin is massively reabsorbed by renal tubules and likely excreted in the biliary tract [63]. (**b**,**c**) Plasma concentration time profiles of CMS (Figure 6b) and formed colistin (Figure 6c) with 105 critically ill patients who were treated by intravenous CMS for blood stream infection or pneumonia caused by multidrug-resistant Gram-negative bacteria (89 not on renal replacement, 12 on intermittent hemodialysis and 4 on continuous renal replacement therapy). Doses of replacement therapy (**d**–**f**). The dashed line indicates the minimum inhibitory concentrations of susceptible strains [69].

**Figure 7 microorganisms-09-01154-f007:**
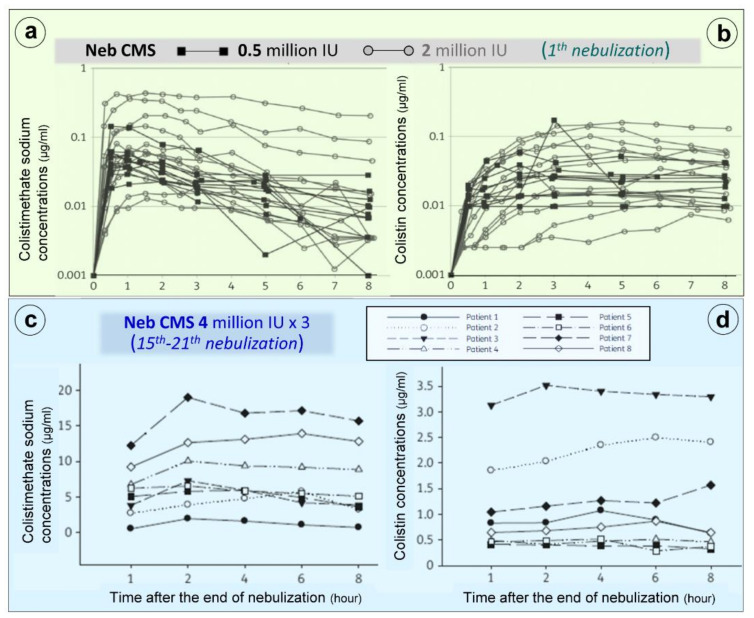
Colistimethate sodium (CMS) and colistin plasma concentrations following nebulization of various doses of CMS. (**a**,**b**) Plasma concentrations measured following the initial nebulization of CMS 0.5 and 2 million IU in a series of twelve patients with ventilator-associated pneumonia caused by Gram-negative bacteria [79]. (**c**,**d**) Plasma concentrations measured after 2–7 days of CMS nebulization at a dose of 4 million IU three times a day in a series of eight patients with ventilator-associated pneumonia caused by multidrug-resistant Gram-negative bacteria. Plasma concentrations were measured by high-performance liquid chromatography in the 8 h following an individual nebulization [81].

**Figure 8 microorganisms-09-01154-f008:**
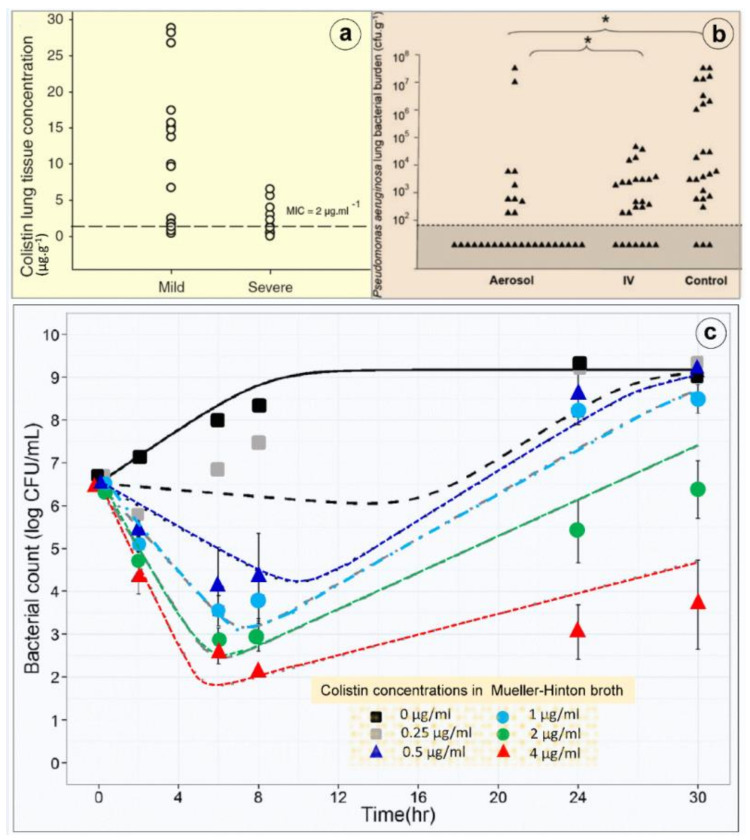
Lung deposition and bactericidal effects of high-dose nebulized colistimethate sodium. (**a**) Colistin concentrations measured in multiple post-mortem subpleural lung specimens in a series of six anesthetized and mechanically ventilated piglets with inoculation pneumonia caused by *Pseudomonas aeruginosa.* Colistin concentrations were measured by high-performance liquid chromatography in 17 pulmonary segments with mild pneumonia and moderate loss of lung aeration and in 13 pulmonary segments with severe pneumonia and complete loss of lung aeration (infectious consolidation). The dashed line indicates the minimal inhibitory concentration of the inoculated *Pseudomonas aeruginosa* [42]. (**b**) Lung bacterial burden of *Pseudomonas aeruginosa* measured in post-mortem lung segments in sixteen piglets with massive inoculation pneumonia caused by *Pseudomonas aeruginosa.* Six received three nebulizations of 100,000 IU/kg colistimethate sodium at 12 h intervals (aerosol), six received four intravenous administrations of 40,000 IU/kg at 8 h intervals (IV) and four did not received any antibiotic (control). Quantitative lung bacteriology was measured in lung segments (triangles) sampled 1 h after the third aerosol in the aerosol group and after the fourth infusion in the intravenous group (IV) and 49 h after the bacterial inoculation in the untreated control group. The grey area indicates the lower limit of quantification for bacterial counts. Asterisk at the top of the figure indicates the statistically significant difference existing between the percentage of lung segments characterized by bacterial counts ranging between 0 and 10^2^ cfu∙g^−1^ in aerosol and intravenous groups and in aerosol and control groups [42]. (**c**) Colistin in vitro time–kill curve. An inoculum of 5 × 10^6^ colony forming unit (CFU)/mL of a wild-type *Pseudomonas aeruginosa* strain was prepared by a suspension of the bacteria from an 18 h logarithmic-growth-phase culture in Mueller–Hinton broth. The experiments were performed in 10 mL glass tubes that were incubated at 37 °C for 18 to 24 h. Colistin was added to obtain concentrations of 0.25, 0.5, 1, 2 and 4 µg/mL (corresponding to 0.5 to eight times the minimal inhibitory concentrations). The bacteria were counted at 0, 2, 6, 8, 24 and 30 h. The limit of quantification was 100 CFU/mL. Four replicates were performed for each concentration. At least one growth control, without added colistin, was included in each experiment. Four replicates were performed for each concentration. Colistin provides a concentration-dependent bacterial killing (means and standard deviations from four replicates and model predicted curves (lines) with mean parameter estimates) [79].

**Figure 9 microorganisms-09-01154-f009:**
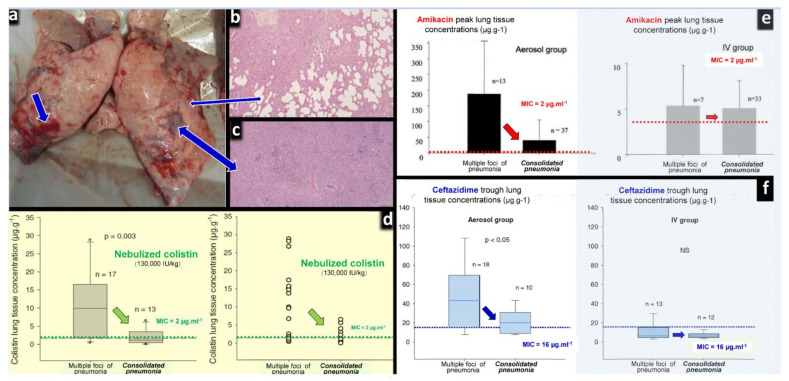
Effects of lung aeration loss on the lung tissue concentrations after the nebulization of high doses of colistin, amikacin and ceftazidime to anesthetized and ventilated piglets with inoculation pneumonia. (**a**) Post-mortem macroscopic view of a piglet’s lungs after intra-bronchial inoculation of *Pseudomonas aeruginosa*. Thick blue arrows indicate lung areas of consolidation and thin blue arrows indicate areas of foci of bronchopneumonia. (**b**) Histologic sections corresponding to foci of bronchopneumonia with persisting lung aeration. (**c**) Histologic sections corresponding to areas of consolidation with complete loss of lung aeration. (**d**) Colistin peak lung tissue concentrations measured by high-performance liquid chromatography 24 h after the intra-bronchial inoculation of *Pseudomonas aeruginosa* (MIC = 2 µg∙mL^−1^) and the nebulization of 130,000 international units∙kg^−1^ of colistimethate sodium (*n* = 6). Following the intravenous administration of high doses of colistimethate sodium (*n* = 6), colistin lung tissue concentrations were undetected. (**e**) Amikacin peak lung tissue concentrations measured by high-performance liquid chromatography 24 h after the intra-bronchial inoculation of *Escherichia coli* (MIC = 4 µg∙mL^−1^), either by nebulization (45 mg∙kg^−1^∙day^−1^, *n* = 10) or by intravenous infusion (15 mg∙kg^−1^∙day^−1^, *n* = 8) [99]. (**f**) Ceftazidime trough lung tissue concentrations measured by high-performance liquid chromatography 24 h after the intra-bronchial inoculation of *Pseudomonas aeruginosa* (MIC = 16 µg∙mL^−1^), and the nebulization of 25 mg∙kg^−1^ at 3 h intervals (*n* = 6) or the continuous intravenous infusion of 90 mg∙kg^−1^∙day^−1^ after an initial rapid infusion of 30 mg∙kg^−1^ (*n* = 6). (Figure 9a–c,f) [42,99,100].

## Data Availability

All relevant data are within the manuscript.

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
