# Peer review of "Nebulized Colistin in Ventilator-Associated Pneumonia and Tracheobronchitis: Historical Background, Pharmacokinetics and Perspectives"

_microorganisms, 2021, doi:10.3390/microorganisms9061154_

Round 1
Reviewer 1 Report
In the manuscript entitled “Nebulized Colistin in Ventilator-Associated Pneumonia and tracheobronchitis: Historical Background, Pharmacokinetics and Perspectives”, Zhu et al reviewed the use of nebulized colistin, in an historical perspective, aiming to provide a historical review of its use in critically ill patients, to describe the pharmacokinetics and pharmacodynamics of intravenous and nebulized colistin, and to suggest future priorities for research of colistin use in nebulization in patients with ventilator associated pneumonia caused by extensive-drug resistant Gram-negative bacteria. In its present form, the work provides an interesting historical perspective of the use of colistin and its importance as a last resource antibiotic, presenting its benefits and limitations based on several studies performed over decades. Moreover, the authors also identify several future directions of research on this thematic, envisaging the extension of this knowledge to the use of other nebulized antibiotics. The manuscript is well organized and in general well written, with some typos and minor corrections being required. A list of issues follows:
Page 2, line 33: “… and is still widely…”
Page 2, line line 48: Please correct the following: “… pneumonias that dis occur were…”
Page 3, line 20: Please correct the following: “…ventilaror-associated…”
Page 3, line 37: Italicize “Pseudomonas aeruginosa”
Page 4, line 3: Italicize “A. baumannii”
Pages 5,6: Figure legend: pay attention to the letter style change.
Page 6, first line after figure 3 legend: correct “Dab are posititively charged…”
Page 7 line 34: rephrase “of respiratory enzymes of the ticarboxylic”, as respiratory enzymes is a vague designation. I suggest the deletion of respiratory enzymes.
Page 12, lines 95-97: the letters indicating each panel should be in bold.
Page 13, 102-103: The sentence seems incomplete: “After intravenous administration of CMS, extensive renal excretion occurs, with some.”
Page 14, line 112: Please check: “… with the permission of the publisher., if intravenous CMS is used, concentrations…”
Page 14, line 138: Is this figure 8 or 7? “(figure 8a-b),”
Page 18, line 243: use VAT instead of “ventilator-associated tracheobronchitis”
Page 22, line 424: “middle and low outcome”. Shouldn´t it be income instead of outcome?
Page 22, line 427: Please correct: “oxacillinase ans ceftolozane tazobactam againsts”
Author Response
- We have made all the corrections suggested.
- Some issues were related to the page layout that mixed the main body of the text with the caption of the figures (page 14 line 112 as an example)
- page 14, line 138 figure 8a,b
Reviewer 2 Report
The review paper by Zhou et al. describes the use of nebulized colistin preparations in medical practice. The review is mostly well written but the manuscript has parts requiring attention:
- There are many typographical errors throughout the manuscript.
- Names of microorganisms have to be written in italics unless you specify a serovar.
- Figure 3 has poor quality and has to be improved.
- Figure 6 shows plasma concentrations vs. time in panels d-f. However, the points are shown on the last part of the curve. What are they? Is this a best fit simulation or a happy design by the authors?
- Figure 7 should be centered.
Author Response
- We careffully reviewed the manuscript and corrected all typographical errors and mispellings
- Names of microorganisms are now in italics. In addition Pseudomonas aeruginosa has been changed for P. aeruginosa and Acinetobacter baummannii for A. baumanni throughout the manuscript
- Figure 3 has 800 dpi and is cannot be improved. It is submitted separately.
- Figure 6d,f is an authentic reproduction and represents a best fit simulation
- Figure 7 is now centered